# FINE-GRAINED GRAPH GENERATION THROUGH LATENT MIXTURE SCHEDULING

## ABSTRACT

Structure aware graph generation aims to generate graphs that satisfy given topological properties. It has applications in domains such as drug discovery, social network modeling, and knowledge graph construction. Unlike existing methods that only provide coarse control over graph properties, we introduce a novel conditional variational autoencoder for fine-grained structural control in graph generation. The approach refines the decoder's latent space by dynamically aligning graph- and property-driven representations to improve both graph fidelity and control satisfaction. Specifically, the approach implements a mixture scheduler that progressively integrates graph and control priors. Experiments on five real-world datasets show the efficacy of the proposed model compared to recent baselines, achieving high generation quality while maintaining high controllability.

## 1 INTRODUCTION

Graph generation is a fundamental task in machine learning for modeling real-world networks such as molecular structures and social networks. Traditional models focus on producing graphs that follow general structural patterns (e.g. power-law distribution in node degree) (Erdös and Rényi, 1959; Barabási and Albert, 1999; You et al., 2018). However, many real world applications require *controlled* graph generation, where generated graphs must satisfy specific topological properties or attributes (Zahirnia et al., 2024; Martinkus et al., 2022). This is particularly crucial in domains such as drug discovery (generating new molecules that satisfy certain chemical properties) (Jin et al., 2018; Shi et al., 2019; Jin et al., 2020; Luo et al., 2021; Popova et al., 2019; Shi et al., 2020; Liu et al., 2021; Zang and Wang, 2020; De Cao and Kipf, 2018), synthetic material design (Wang et al., 2022; Sanchez-Lengeling and Aspuru-Guzik, 2018), social and information networks (Pitas, 2016; Zhou et al., 2020; Zeno et al., 2021), knowledge graphs (Melnyk et al., 2022; Zhou et al., 2023; Cao et al., 2023), and programming languages (generating program graphs from source codes) (Allamanis et al., 2018).

Despite significant progress in graph generation, existing controlled graph generation methods are limited in scope, often restricting control to basic graph attributes such as node and edge counts as opposed to more fine-grained structural constraints, and lack a principled way to balance the structural generation process and attribute-based constraints. For example, EDGE Chen et al. (2023) is a discrete diffusion model that explicitly focuses on node degrees to control graph generation; DiGress Vignac et al. (2023) also builds on discrete diffusion techniques to incorporate properties such as planarity or acyclicity for generating graphs; Spectre Martinkus et al. (2022) is a generative adversarial network that control graph generation by focusing on eigenvalues and eigenvectors, which provide abstract control over topological properties; and GenStat Zahirnia et al. (2024) is a variational autoencoder that learns a latent adjacency matrix from attributes such as number of edges, triangles, and $k$-hop neighbors histogram. Other works such as (Yang et al., 2019; Ommi et al., 2022) uses class labels and other class information as a condition to generate graphs where as (Liu et al., 2024; Mercatali et al., 2024) focuses on molecule generation tasks.

We propose TOPOGEN, a novel conditional variational autoencoder for controlled graph generation based on fine-grained topological attributes. TOPOGEN uses both the adjacency matrix and desired topological attributes during training for better latent space alignment and improved decoder tuning, *while relying only on attributes during inference*. We propose a novel scheduling mechanism (MIXTURE-SCHEDULER) that progressively integrates structure- and attribute-driven latent repre-

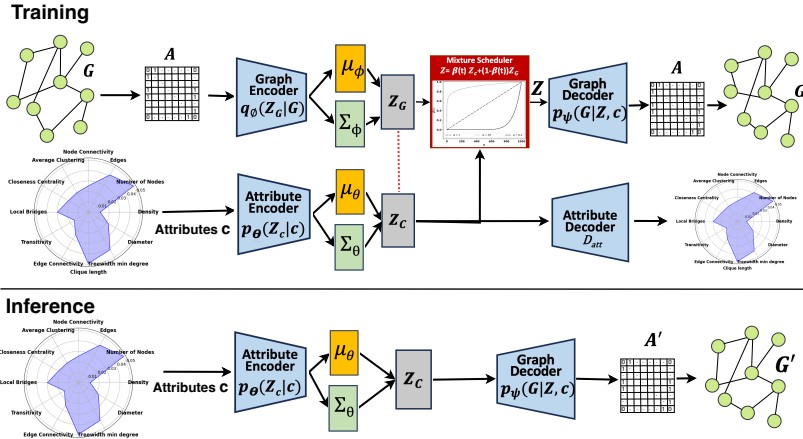

Figure 1: TOPOGEN uses both graph attributes and adjacency matrix *during* training for improved decoder tuning. It implements a novel scheduling technique to effectively integrate attributes and graph distributions to provide fine-grained topological control in generation. At test and inference times, it only relies on desired attributes to generate graphs.

sentations for adaptive and precise control over generated graphs. TOPOGEN provides flexibility in graph generation by generalizing to any number of fine-grained control attributes.

The contributions of this work are:

- TOPOGEN: a novel conditional variational autoencoder that enables fine-grained topological control using both graph adjacency matrices and attribute vectors during training, while relying only on attributes during inference for precise graph generation.
- MIXTURE-SCHEDULER: a latent space integration technique to dynamically balance adjacency matrix and attribute representations for generation.

We compare TOPOGEN against current models for controlled graph generation on several datasets and generation tasks. Our key findings are as follows: (1) Joint adjacency-attribute integration improves generation quality, aligning graphs with specific attribute constraints more effectively than prior models; (2) Gradual incorporation of prior information (from attribute via MIXTURE-SCHEDULER) during training improves controlled graph generation; (3) Increasing the number of control attributes improves generation precision, which confirms that fine-grained constraints help generate structurally valid graphs. Our code and data will be released.

## 2 CONTROLLED GRAPH GENERATION

**Problem Definition** Given a vector $\mathbf{c}$ that represents the fine-grained topological attributes of a target graph $G$, our goal is to generate a graph $\hat{G}$ whose structure satisfies the attributes $\mathbf{c}$.

**Solution Overview** We formulated the above problem as a "learning to generate" task. During *training*, TOPOGEN uses the adjacency matrix $\mathbf{A}$ of the target graph $G = (V, E)$ and its corresponding attribute vector $\mathbf{c}$ to learn the joint distribution of graphs and their attribute vectors for controlled graph generation. As Figure 1 shows, TOPOGEN encodes the structural representation $\mathbf{Z_G}$ from adjacency matrix $\mathbf{A}$ to parameterize the posterior distribution $q_\phi$, and the attribute representation $\mathbf{Z_c}$ from the attribute vector $\mathbf{c}$ to define the prior distribution $p_\theta$. These representations are combined using MIXTURE-SCHEDULER to balance structural and attribute information in the latent representation $\mathbf{Z}$. The MIXTURE-SCHEDULER aims to align $q_\phi$ and $p_\theta$, as they both represent graphs with the same topological structure. The decoder then learns the likelihood distribution $p_\psi$ from $\mathbf{Z}$ to generate a graph $\hat{G}$ that satisfies the specified attributes $\mathbf{c}$. At *inference* time, TOPOGEN generates graphs using only the prior $p_\theta$ and the likelihood $p_\psi$, conditioned solely on the attribute vector $\mathbf{c}$.

**Control Attributes**   We provide a list of structural attributes for explicit and precise control over the graph generation process. These include **number of nodes & edges**, which define the scale of the target graph; **number of local bridges**, which is the number of edges that are not part of a triangle in the graph, these "bridges" transfer information between different graph regions; **graph density**, which is the fraction of possible edges in the graph, computed as $e/v(v-1)$, where $e$ is the number of edges and $v$ is the number of nodes in the graph; **edge connectivity**, which is the minimum number of edges that must be removed to disconnect the graph; **node connectivity**, which is the minimum number of nodes that must be removed to disconnect the graph; **number of maximum cliques**, which is the number of maximal complete subgraphs in the graph; **graph diameter**, which is the length of the shortest path between the most distanced nodes in a graph; **treewidth min degree**, which is an integer quantifying how much the graph deviates from a tree; **closeness centrality**, which is the average distance of a node to all other nodes in its corresponding connected component, averaged across all nodes; **clustering coefficient**, which is the fraction of triangles within a node's immediate neighbors, averaged across all nodes; and **transitivity**, which is the fraction of all possible triangles present in a graph, computed as $3 \times |triangles|/|triads|$, where a "triad" is a set of three nodes connected by at least two edges.

**Importance:**   These attributes enable precise control over graph generation and make it possible to generate graphs that satisfy diverse and complex structural requirements. Attributes such as transitivity and graph density can be adjusted to manage the connectivity of graphs. For example, increasing graph density can improve the robustness of local area networks in terms of reliable communication, fine-tuning transitivity can help model molecular structures with specific bonding properties, or adjusting transitivity can help simulate disease spread patterns in human contact networks in epidemiology. Granular control over these attributes allows for generating graphs that satisfy specific needs across various applications. This includes creating balanced graph datasets with controlled structural diversity; augmenting small-scale datasets by generating similar yet distinct subgraphs, especially in fields like medicine, where obtaining large-scale real-world data is expensive or infeasible; and finding novel structures in fields such as chemistry and molecular biology. For example, in drug discovery, the generation of graphs that represent potential compounds with desired properties can accelerate the search for and discovery of new therapeutics.

## 2.1 Graph Encoding through Mixture Scheduling

We introduce a new approach to graph encoding by gradually balancing structural (i.e. adjacency matrix) and attribute-based representations during training. Unlike conventional methods that rely on direct sampling or divergence minimization (e.g., Wasserstein distance or KL divergence), our approach dynamically controls the contribution of structural and attribute representations using a smooth scheduling function. This allows for flexible and adaptive representation learning, where generated graphs preserve topological properties as well as align with desired attribute constraints.

**Graph Encoder**   TOPOGEN's encoder uses a convolution neural network (CNN) to encode the structural information of graph $G$ into a latent representation $\mathbf{Z_G}$ using $s$ channels, and parameterize the posterior distribution $q_\phi$. This distribution is defined as:

$$q_\phi\left(\mathbf{Z_G}|G\right) \quad = \quad \mathcal{N}\left(\mathbf{Z_G}|\mu_\phi = h\left(G\right), \Sigma_\phi = h'\left(G\right)\right), \tag{1}$$

where $\mathcal{N}$ is a Gaussian distribution with mean vector $\mu_\phi = h\left(G\right)$ and covariance matrix $\Sigma_\phi = h'\left(G\right)$, both obtained from the CNN with parameters $\phi$, where $h(G)$ computes the mean vector using features from the first half $(s/2)$ of CNN channels and $h'(G)$ computes covariance matrix from the second half $(s/2)$ of the CNN channels. The partitioning is similar to how variational autoencoders (VAEs) separate their latent space into a mean and variance to generate diverse samples while preserving meaningful structure. It allows different parts of the CNN to capture distinct statistical properties of the latent space, explicitly control uncertainty and variability, and encode well-structured representations. While TOPOGEN is compatible with GNNs, we focus on CNNs due to better performance in our experiments. Other frameworks have used Multilayer Perceptron (MLP) layers Zahirnia et al. (2024); Vignac et al. (2023); Jo et al. (2024) or a combination of LSTM, MLP and message passing Chen et al. (2023) as graph encoder.

**Attribute Encoder** To control graph generation, the attribute encoder in Figure 1 learns the representation of the attribute $\mathbf{c}$, and the parameters for the prior distribution $p_\theta$ are learned as follows:

$$p_\theta\left(\mathbf{Z_c}|\mathbf{c}\right) \;\; = \;\; \mathcal{N}\left(\mathbf{Z_c}|\mu_\theta = f\left(\mathbf{c}\right), \Sigma_\theta = I\right), \tag{2}$$

where $f(\mathbf{c})$ is a non-linear transformation of the attribute vector from a feed forward neural network to capture interaction between the features of the attribute representation, and $\Sigma_\theta$ is the unit variance.

**Mixture Scheduler** Unlike conventional approaches that align prior and posterior distributions using Wasserstein distance (Kantorovich, 1960) or divergence techniques (Kullback and Leibler, 1951), we introduce MIXTURE-SCHEDULER, a principled approach that gradually integrates the prior $p_\theta$ and posterior $q_\phi$ to learn effective representations that satisfy desired attribute $\mathbf{c}$. Instead of abrupt transitions, MIXTURE-SCHEDULER enables a smooth and adaptive interpolation between structural and attribute-based latent representations. We define the final latent representation as:

$$\mathbf{Z} = \beta(t)\mathbf{Z_c} + (1 - \beta(t))\mathbf{Z_G}, \tag{3}$$

where $\beta(t)$ is the *inclusion factor* at epoch $t$, which controls the gradual incorporation of the prior $\mathbf{Z_c}$ during training. To derive a general form of $\beta(t)$, we assume that the rate by which the prior $\mathbf{Z_c}$ is incorporated is uniformly distributed over the remaining training time:

$$\frac{d\beta(t)}{dt} = \frac{1 - \beta(t)}{1 - t}, \tag{4}$$

where $t \in [0, 1]$ is normalized training progress, with $t = 1$ when $\mathbf{Z_c}$ is fully incorporated. Solving this differential Equation, we obtain:

$$\int \frac{1}{1 - \beta(t)} d\beta(t) = \int \frac{1}{1 - t} d(t), \tag{5}$$

which results in $\beta(t) = 1 - \exp(c)(1 - t)$ for some constant $c$. Setting the initial inclusion value as $\beta(0)$ (at $t = 0$) and $\beta(1) = 1$, we obtain a *linear* scheduler:

$$\beta(t) = \min\left(1, 1 - (1 - \beta(0))(1 - t)\right). \tag{6}$$

We modify the linear scheduler to allow for adaptive control over the rate at which $\mathbf{Z_c}$ is incorporated at different training stages. This results in the *generalized inclusion function*:

$$\beta(t) = \min\left(\gamma, (1 - (1 - \beta(0))(1 - t))^{\frac{1}{\alpha}}\right), \tag{7}$$

where $\gamma \in [0, 1]$ controls the maximum possible inclusion from prior $p_\theta$; $\alpha > 0$ determines the rate at which the prior is integrated during training, see Figure 2; $t$ represents the current epoch; and $\beta(0)$ is the initial inclusion value. The intuition behind developing (7) is to provide flexible control over the contributions of the prior and posterior and allow for smooth and gradual transition between them; see Figure 2. By gradually increasing the influence of $\mathbf{Z_c}$, the learned representations retain meaningful graph topology while aligning with the desired attribute constraints.

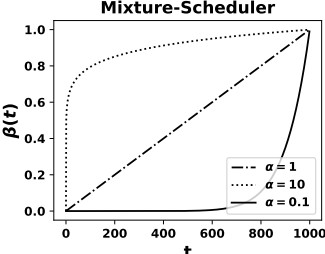

Figure 2: The parameter $\alpha$ controls the inclusion factor, $(\beta(t))$ in (3). It specifies how quickly the prior is integrated during training. A smaller $\alpha$ results in less inclusion of $p_\theta$ during the early training epochs, with gradual inclusion increasing toward the end of training.

The MIXTURE-SCHEDULER can be understood as a *soft optimization constraint* that enables smooth interpolation between probability distributions. It dynamically transitions the latent representation $\mathbf{Z}$ from the structural posterior $q_\phi(\mathbf{Z_G}|G)$ to the attribute-conditioned prior $p_\theta(\mathbf{Z_c}|\mathbf{c})$ during training. The scheduler implicitly minimizes the Wasserstein distance ($W$) between the structural and attribute-driven distributions controlled by the scheduling parameters $\alpha$ and $\gamma$, to govern the transition dynamics: a smaller $\alpha$ results in a slow transition, i.e. prioritizing structural learning before enforcing attribute constraints; a larger $\alpha$ causes a faster shift towards $\mathbf{Z_c}$, i.e. aligning graphs with attributes earlier but risking instability; and $\gamma$ controls the final alignment of $\mathbf{Z}$ with $\mathbf{Z_c}$. Higher $\gamma$ enforces stronger attribute constraints but may distort structural properties.

## 2.2 ATTRIBUTE-GUIDED GRAPH GENERATION

TOPOGEN introduces a novel attribute-guided graph generation framework. Unlike conventional methods that rely solely on structural embeddings, our framework incorporates graph attributes as constraints to enable precise controlled graph generation. Using a Bernoulli-based likelihood model, we allow flexible edge prediction while maintaining topological consistency. In addition, we introduce a distance-regularized objective function to enforce smooth transitions between prior and posterior distributions to balance structural fidelity with attribute adherence.

**Graph Generation** We model graph generation using a Bernoulli distribution Murphy (2012) to determine edge probabilities between node pairs and generate the adjacency matrix $\mathbf{A}$. The graph decoder learns the likelihood distribution $p_\psi$ from $\mathbf{Z}$ to maximize the probability of generating graphs that satisfy the attribute constraints $\mathbf{c}$:

$$p_\psi\left(G|\mathbf{Z}, \mathbf{c}\right) \sim \text{Bernoulli}\left(D_{graph}\left(\mathbf{Z}\right)\right), \tag{8}$$

where $\mathbf{Z}$ represents the latent representation processed by the decoder $D_{graph}$ to obtain the parameters of the Bernoulli distribution. Here, a value of 1 from the Bernoulli distribution indicates an edge between a node pair.

**Training Objective** We develop the following objective function to learn model parameters:

$$\mathcal{L}(\phi, \theta, \psi|G, \mathbf{c}) = \tag{9}$$

$$\underbrace{\mathbb{E}_{q_\phi(\mathbf{Z}|G)}\left[\log p_\psi(G|\mathbf{Z}, \mathbf{c})\right]}_{graph-reconstruction} - \lambda_d \cdot \underbrace{\mathcal{D}\left(q_\phi(\mathbf{Z_G}|G), p_\theta(\mathbf{Z_c}|\mathbf{c})\right)}_{distance-function} + \lambda_c \cdot \underbrace{\mathbb{E}_{p_\theta(Z_c|\mathbf{c})}\left[(\mathbf{c} - D_{\text{att}}(Z_c))^2\right]}_{attribute-reconstruction},$$

where the **first term** is the reconstruction loss, which encourages generating graphs that are structurally similar to the given graph $G$, conditioned on the latent representation $\mathbf{Z}$ and attributes $\mathbf{c}$. The **second term** ($\mathcal{D}$) is a general distance function for probability distributions; it regularizes the objective by computing the difference between the approximate posterior $q_\phi(\mathbf{z}|G)$ and the prior $p_\theta(\mathbf{z}|\mathbf{c})$ to explicitly enforce alignment between learned graph structures and attribute-driven representations. We note that MIXTURE-SCHEDULER implicitly aligns the posterior ($\mathbf{Z_G}$) and prior ($\mathbf{Z_c}$). Without explicit regularization, $\mathbf{Z_G}$ and $\mathbf{Z_c}$ may remain disjoint, and result in poor attribute-guided graph generation. The second term provides an explicit constraint for smoother transitions, prevents posterior drift, and stabilizes training by enforcing gradual alignment between structural and attribute-driven latent spaces. The distance function, $\mathcal{D}$, can be chosen based on application needs. We used the Wasserstein distance due to its symmetric property. However, other distance functions can also be used. The **third term** encourages accurate reconstruction of the attribute vector $\mathbf{c}$, using a neural network based attribute decoder $D_{att}()$, described below. $\lambda_d$ and $\lambda_c$ are hyperparameters to balance these terms.

**Attribute Decoder** During training, we use a feedforward neural network as the attribute decoder to reconstruct attributes from latent representation $\mathbf{Z_c}$. To guide accurate graph generation that aligns with the specified control attributes, we minimize the mean square error (MSE) between the ground truth and predicted attribute vectors and effectively guide the model toward attribute-consistent generation; see the third term in (9).

**Inference Process** During inference, the model generates a graph conditioned on the desired attribute vector $\mathbf{c}$ using the prior distribution $p_\theta$, as illustrated in Figure 1. The prior is first used to create a latent representation, which encodes attribute-driven structural properties. This representation is then passed to the decoder to parameterize the $p_\psi$ distribution to sample and generate a graph that satisfies the specified attributes. Unlike training, inference relies only on the prior, so that graph generation is fully controlled by the desired attributes without requiring reference graphs.

## 3 EXPERIMENTS

**Datasets** We use several datasets for experiments: **WordNet** (Miller, 1995): a large lexical dataset of English, where words are grouped into synonym groups (synsets) and are connected by linguistic

relationships. We construct four distinct WordNet graphs using hypernyms, hyponyms, meronyms, and holonyms relations. **Ogbn-arxiv** (Hu et al., 2020): The Open Graph Benchmark dataset includes a citation network of computer science papers from arXiv, with nodes as papers and edges represent citations among papers. Each paper carries an embedding derived from its title and abstract. **Citeseer** (Kipf and Welling, 2017): a citation network of scientific articles, where nodes are papers and edges indicate citations between them. **MUTAG** (Morris et al., 2020): a molecular dataset where each graph represents a chemical compound labeled based on its mutagenic effect on specific gram negative bacterium. **MOLBACE** (Hu et al., 2020): a molecular dataset where each graph represents a chemical compound. We create several datasets of graphs by extracting $k$-hop neighbors, $k = \{2, 3\}$, around each node in the above graphs to create training, validation and test data splits for controlled graph generation. Table 1 shows the statistics of these datasets.

Table 1: Dataset statistics in terms of number of graphs.

| Dataset | Train | Val | Test |
|---------|-------|-----|------|
| **WordNet** | 52,675 | 2,926 | 2,927 |
| **Citeseer** | 1,406 | 78 | 79 |
| **Arxiv** | 47,538 | 2,641 | 2,641 |
| **MUTAG** | 169 | 10 | 9 |
| **MOLBACE** | 1,323 | 74 | 74 |

**Evaluation Metrics** We compute the difference between predicted and ground truth graphs to compare models in controlled graph generation using two metrics: **Graph Edit Distance (GED↓)** Sanfeliu and Fu (1983): is a structural similarity (or dissimilarity) measure that quantifies the minimum number of edit operations (node/edge insertions, deletions, or substitutions) required to transform one graph into another. It provides a fine-grained comparison by explicitly capturing structural differences. However, GED is computationally expensive, as finding the exact edit distance between two graphs is NP-hard Zeng et al. (2009). Therefore, it is often used to determine structural similarity among small graphs. **Spectral Difference (SD↓)** (Jo et al., 2024): a widely used approach for comparing structural properties of graphs. It uses the sorted eigenvalues of the Laplacian matrix, which encode global structural properties such as connectivity, clustering tendencies, and diffusion dynamics. Unlike node-to-node matching methods like GED, SD is invariant to node ordering, robust to small local perturbations, and computationally efficient. For fair and meaningful comparison between predicted and ground truth graphs of different sizes, we align their eigenvalue ($\lambda$) distributions by zero-padding the smaller graph's eigenvalues to match the size of the larger graph, and report average spectral difference, $SD = 1/n \times ||\lambda_{groundtruth} - \lambda_{pred}||_2$ for each dataset. We chose SD and GED as evaluation metrics to specifically focus on fine-grained attribute fidelity and node-to-node alignment, which are most directly influenced by different models' attribute-conditioning objectives. **Maximum Mean Discrepancy (MMD↓)** (You et al., 2018; Vignac et al., 2023) is widely used for distribution-level comparison; however, they do not provide fine-grained evaluation on individual attributes. MMD results are reported in Appendix.

**Baselines** We compare TOPOGEN against several recent baselines. For a fair comparison, we incorporate our control attributes into all models that support conditioning. The exception is GraphRNN, which is a free (non-controlled) generative model and provides a point of comparison for evaluating the benefits of attribute conditioning. **GraphRNN** (You et al., 2018): generates graph iteratively by training on a representative set of graphs using breath first search of nodes and edges and implements node and edge RNNs to generate target graphs. GraphRNN is not a controlled generation approach. **EDGE** (Chen et al., 2023): is a diffusion based generative model which iteratively removes edges to create a completely disconnected graph and uses decoder to iteratively reconstruct the original graph. It explicitly uses adjacency matrix to satisfy the statistics of the generated graphs during training. **GenStat** (Zahirnia et al., 2024): learns the latent adjacency matrix conditioned on graph level attributes, and decodes it to recreate attribute statistics and use them to generate graphs. **DiGress** (Vignac et al., 2023): learns to generate graphs by discrete denoising diffusion model with categorical nodes and edge attributes and by incorporating graph-theoretic features. **GruM** (Jo et al., 2024): is a graph generation framework which captures the topology of the graph and predicts the graph using mixture of endpoint-conditioned diffusion processes.

## 3.1 MAIN RESULTS

Table 2 shows the overall performance of models across datasets. TOPOGEN consistently achieves the lowest SD across all datasets and the lowest GED across 3 out of 5 datasets, which indicate more accurate structural alignment and better controlled graph generation. DiGress achieves the

Table 2: Performance comparison across multiple datasets. We evaluate models using Spectral Difference (SD) (left) and Graph Edit Distance (GED) (right), where lower scores indicate better performance. All models are optimized using the same set of attribute constraints described in §2.

| | WordNet | Citeseer | Ogbn-Arxiv | MUTAG | MOLBACE | WordNet | Citeseer | Ogbn-Arxiv | MUTAG | MOLBACE |
|---|---|---|---|---|---|---|---|---|---|---|
| | SD ($\downarrow$) | | | | | GED ($\downarrow$) | | | | |
| **GraphRNN** | **0.31** | 0.42 | 0.46 | 0.21 | 0.13 | 32.58 | 54.83 | 52.43 | 15.77 | 41.52 |
| **GenStat** | 0.32 | 0.40 | 0.47 | 0.21 | 0.11 | 37.01 | 58.69 | 63.77 | 34.66 | 61.32 |
| **EDGE** | 0.32 | 0.44 | 0.47 | 0.27 | 0.14 | 35.16 | 59.58 | 59.70 | 29.77 | 72.93 |
| **DiGress** | 0.37 | 0.85 | 0.73 | 0.88 | 0.91 | 31.32 | **45.16** | **47.24** | 32.66 | 61.64 |
| **GruM** | 0.40 | 0.43 | 0.50 | 0.52 | 0.69 | 29.68 | 55.06 | 51.87 | 27.00 | 65.27 |
| **TOPOGEN** | 0.44 | **0.27** | **0.40** | **0.10** | **0.09** | **26.79** | 45.62 | 49.68 | **12.88** | **28.90** |

Table 3: Graph visualization across datasets. Examples are taken from test splits of datasets.

best GED on Citeseer and Ogbn-Arxiv, but struggles in SD and underperforms on domain specific datasets like MUTAG and MOLBACE. GruM has competitive SD scores in WordNet, Citeseer, and Ogbn-Arxiv, but performs worse in GED.

The high GED performance of DiGress on citation networks (Citeseer and Ogbn-Arxiv) is perhaps because DiGress is optimized for handling dense and scale-free networks, whereas TOPOGEN is designed for fine-grained attribute control and may not explicitly prioritize preserving connectivity hubs. In addition, DiGress likely preserves local citation patterns better than TOPOGEN, leading to lower GED scores. In addition, GraphRNN achieves lower SD and GED scores on MUTAG and MOLBACE compared to most baselines. Unlike citation or social networks, molecular graphs have high local dependencies–atoms must be connected in precise ways to form valid molecules, where certain structures appear frequently (e.g., benzene rings, carbon chains). The sequential approach of GraphRNN perhaps better learns these recurring patterns, which makes it effective for generating realistic molecular graphs. Other baselines (DiGress and GruM) underperform on capturing the fine-grained rules that govern molecular connectivity.

Table 3 shows examples of different graphs generated by TOPOGEN and GruM across datasets; see Appendix 5.4, Table 6 for outputs of other models. As evident from the Table, TOPOGEN generates graph that are more similar to the target graphs compared to other baseline models. We attribute this improvement to TOPOGEN's ability to perform fine-grained controlled generation using graph attributes. Generation error for each attribute is detailed in Appendix 5.5.

## 3.2 MODEL INTROSPECTION

We conduct several ablation studies to understand the effectiveness of TOPOGEN in controlled graph generation. We analyze scalability to larger number of nodes; provide insights on generating graphs by masking fundamental attributes like number of nodes and edges, while providing all other fine-grained attributes; and provide a detailed study on MIXTURE-SCHEDULER, to analyze the effects of limiting the inclusion factor and varying the rate of inclusion. In addition, we conduct ablation study of MIXTURE-SCHEDULER to answer following questions: (RQ1) Does including the prior distribution $p_\theta$ help? (RQ2) How does the rate of inclusion affect model's performance? (RQ3) How much of the prior should be included?

**Contribution of control attributes** Graph attributes determine the required structural properties of generated graphs. Figure 3 shows the effect of independently removing one at-

Table 4: Effect of removing individual components from the objective function of TOPOGEN. Bold indicates the highest error, marking the most influential component.

| | WordNet | Citeseer | Ogbn-Arxiv | MUTAG | MOLBACE |
|---|---|---|---|---|---|
| | SD($\downarrow$) | | | | |
| TOPOGEN w GNN as Graph encoder | 0.32 | 0.50 | 0.57 | 0.18 | 0.15 |
| TOPOGEN w/o Distance Function | 0.69 | 0.45 | **1.36** | **0.49** | **0.99** |
| TOPOGEN w/o Attribute Reconstruction | 1.12 | 0.28 | 0.40 | 0.14 | 0.15 |
| TOPOGEN w/o MIXTURE-SCHEDULER | **4.66** | 0.26 | 0.40 | 0.16 | 0.13 |
| TOPOGEN w/o Adjacency matrices during training | 0.62 | **0.53** | 0.55 | 0.17 | 0.17 |
| **TOPOGEN** | 0.44 | 0.27 | 0.40 | 0.10 | 0.09 |

tribute at a time for each training run. Removing either density, closeness centrality or transitivity results in increase in error compared to the number of nodes, average clustering or number of local bridges. This suggests that TOPOGEN learns more detailed structural patterns and generates more accurate graphs when guided by a more set of attributes.

In fact, introducing more restrictive constraints than basic attributes–those such as density or closeness centrality–further refines the generation process and results in graphs that better preserve the intended structural properties. Here, NC (node connectivity), EC (edge connectivity), TWMD (tree width min degree), Avg Clust (average clustering), LB (number of local bridge), Clique (number of cliques).

Figure 3: Plot shows increase in generation error of the specific attribute when not included in training. Blue line indicates including all attributes.

**Generation without number of nodes and edges**
Figure 4(a) compares the performance of TOPOGEN when trained with and without the number of nodes and edges as explicit control attributes. The results show that the model achieves similar performance even without these basic attributes, which suggests that TOPOGEN can infer the number of nodes and edges with minimal error using other fine-grained structural attributes. This demonstrates the model's ability to capture graph properties and generate structurally consistent graphs without relying on direct node and edge count supervision, which are commonly used by other models.

**Contribution Analysis of components from TOPOGEN**   Table 4 reports an ablation study of the TOPOGEN objective function across five datasets highlighting the effective components. In WordNet, removing the MIXTURE-SCHEDULER causes a sharp error increase (4.66 vs. 0.44), making it the most critical, followed by attribute reconstruction (1.12). In Citeseer, using adjacency matrix increases error to (0.53) and use of GNN as a graph encoder to (0.50), underscoring the role of both attribute learning and type of structural encoding. For Ogbn-Arxiv, GNN as a graph encoder increases the error to (0.57) and the distance function to (1.36) which is the key for aligning prior $p_\theta$ and posterior $q_\phi$ distributions. In Mutag and Molbace, the distance function (0.49, 0.99) and using adjacency matrices (0.17, 0.17) guide TOPOGEN towards lower error. Finally, replacing CNN with a GNN encoder Xu et al. (2019) consistently degrades performance. We hypothesize that this is due to over-smoothing, making GNN struggle to precisely reconstruct the graph structures. These results confirm that each component contributes to the reduction of the generation error, with MIXTURE-SCHEDULER and the distance function being the most influential overall.

**RQ1: Does including the prior distribution $p_\theta$ help?**   We consider three scenarios: (i) when the model only learns from $q_\phi$ distribution ($\beta(t) = 0$), (ii) when the model gradually combine $p_\theta$ and $q_\phi$ as training progresses ($\beta(t) \to \gamma$), and (iii) when the model combines both $p_\theta$ and $q_\phi$ with constant influence factor $\beta(t) = \gamma$. As shown in Figure 4(b), combining representations from both distributions $p_\theta$ and $q_\phi$ helps generate better graphs compared to using only representations from $q_\phi$. Also, gradual increase in influence factor $\beta(t) \to \gamma$ performs better compared to keeping

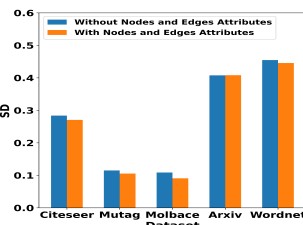
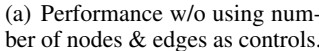

(a) Performance w/o using number of nodes & edges as controls.

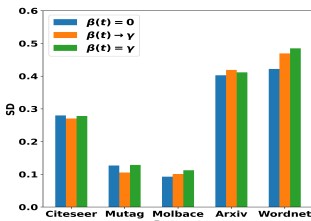

(b) Effect of inclusion factors on generation error across datasets.

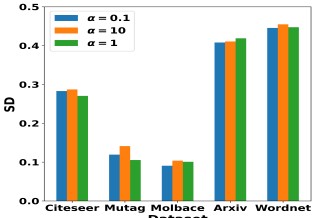

(c) Effect of $\alpha$ on error in generating graphs across datasets.

Figure 4: Ablation Analysis

it constant $\beta(t)$. We conclude relying only on graph representation from $q_\phi$ without considering attribute representation from $p_\theta$ results in higher SD error and lower performance.

**RQ2: How does the rate of inclusion affect model's performance?** We analyze different rates of inclusion. As Figure 4(c) shows, a slow inclusion rate ($\alpha = 0.1$) often helps model in learning better representations compared faster inclusion rates, e.g. ($\alpha$=10). This result suggests that initially focusing on the $q_\phi$ and gradually incorporating the $p_\theta$ yields better latent representations.

**RQ3: How much of the prior should be included?** We vary the influence of prior distribution by adjusting the maximum inclusion rate, $\gamma \in [0, 1]$, where $\gamma =0$ excludes $p_\theta$ entirely, and $\gamma = 1$ excludes the posterior $q_\phi$. Figure 5 shows that smaller values of $\gamma$ result in lower error, suggesting that limited inclusion of $p_\theta$ improves graph generation by better balancing both distributions.

### 3.3 DE-NOISING GRAPH ATTRIBUTES

We evaluate TOPOGEN's robustness to noisy attributes by masking one attribute at a time during inference. Using the best trained model with frozen parameters, we run 12 inference passes, each time setting one attribute to zero across all test graphs. Figure 6 shows the results, with the dotted line as the baseline SD error without masking. TOPOGEN remains resilient, often generating accurate graphs despite missing controls. The largest error increases occur when edges, local bridges, or cliques are masked, confirming their critical role in structural fidelity, whereas masking clustering coefficient, transitivity, or diameter yields only minor changes, indicating they refine finer structural details.

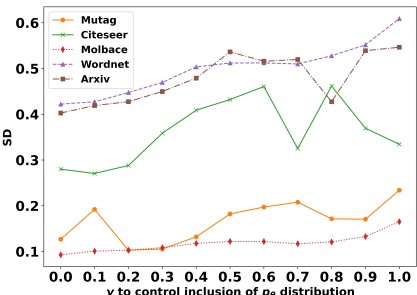

Figure 5: Relation between the maximum inclusion rate $\gamma$ and SD error. MIXTURE-SCHEDULER reduces SD error by combining information from both distributions.

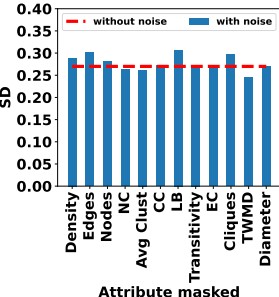

Figure 6: Bars indicate SD on Citeseer with one attribute masked (set to zero) at a time. The dotted line marks TOPOGEN's performance without masking.

## 4 CONCLUSION AND FUTURE WORK

We presented TOPOGEN, a novel controlled graph generation model that generates graphs satisfying fine-grained topological attributes. It includes a novel distribution scheduler, MIXTURE-SCHEDULER, to combines *attribute* and *adjacency matrix* representations for learning accurate latent structures. TOPOGEN enables precise control–even without explicitly specifying basic properties such as node and edge counts–and achieves lower generation error by gradually integrating multiple control attributes. In future, we plan to extend TOPOGEN to dynamic or temporal graphs for applications such as in social network analysis, traffic prediction, and temporal knowledge graphs.

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

# 5 APPENDIX

## 5.1 SCALABILITY TO FINE-GRAINED CONDITIONALLY GENERATE LARGE GRAPHS

We analyze the effect of increasing the maximum number of nodes, $|V|$, on TOPOGEN's SD performance. Table 5 shows that SD increases as the maximum number of nodes grows up to 200 nodes. This is because larger graphs have greater structural complexity, with more potential edges and relationships that are harder to generate accurately. This makes it challenging for the model to capture both local and global topological properties, and potentially leads to cumulative errors in matching

node-specific attributes such as degrees and centrality. In addition, larger graphs often contain more variability and sparsity, which further complicates satisfying the desired structural attributes and result in higher deviations between the generated and target graphs.

Table 5: Generation performance degrades as the target number of nodes increases.

| #Nodes | SD($\downarrow$) |
|---|---|
| 50 | 0.40 |
| 60 | 0.60 |
| 80 | 0.74 |
| 100 | 0.82 |
| 200 | 0.83 |

## 5.2 LIMITATION

TOPOGEN performs significantly well compared to recent baselines to generate graphs from given fine-grained control attributes. However, the performance degrades as we aim to generate larger graphs because of two factors: increased complexity in aligning multiple fine-grained attribute constraints as graph size grows, and the use of less expressive encoders (such as convolutional encoders) for capturing long-range dependencies in large graphs.

## 5.3 SETTINGS

Following previous works (De Cao and Kipf, 2018; Zahirnia et al., 2024), we set the maximum number of nodes to $V = 50$ in experiments. This threshold is appropriate, given the common practice of sampling 1-2 hop subgraphs for nodes. We set the number of hops to $k = 2$ for all datasets except for Citeseer, for which we use $k = 3$ due to its smaller size. In addition, we extract graph attributes using Networkx (Hagberg et al., 2008). We consider a maximum number of 1000 training iterations for Citeseer and 200 iterations for other datasets, which is sufficiently large for convergence. For the CNN encoder, we use two convolutional layers with kernel size of 5, and 32 and 64 channels respectively for all datasets. For the decoder, we used two convolutional layers with 64,32 channels respectively. The model requires approximately 260M FLOPs for the graph encoder, 260M for the graph decoder, 231M for the attribute encoder, and 231M for the attribute decoder. In total, it contains about 116M trainable parameters, consisting of roughly 100k from the CNN components and 115M from the MLP with a hidden dimension size of 1024. For hyperparameters, we set the maximum possible inclusion from prior $p_\theta$ ($\gamma$) to 0.3 for Mutag, 0.1 for Molbace, Citeseer, and arxiv; and 0.2 for Wordnet. We consider a batch-size of 1,028 and run all our experiments on a single A100 40GB GPU.

## 5.4 GRAPH VISUALIZATION

Table 6 shows examples of different graphs generated by baselines across datasets.

## 5.5 ATTRIBUTE-WISE ERROR ANALYSIS

Table 7 shows the absolute mean error between the ground truth value and predicted value for each attribute. We use mean absolute difference (MAD$\downarrow$) metric for evaluation. MAD computes the absolute difference between the attributes of predicted graphs and their corresponding target graphs. We average these differences for each dataset. Abbreviations NC (node connectivity), EC (edge connectivity), TWMD (tree width min degree), Avg Clust (average clustering), LB (number of local bridge), Clique (number of cliques).

## 5.6 DIFFERENT GRAPH ENCODER

To study the importance of using different graph encoders, we use Graph Neural Network (GNN) Xu et al. (2019) instead of CNN in TOPOGEN. As shown in Table 11, MAD error increases.

Table 6: Graph visualization across datasets. Examples are taken from test splits of datasets.

| | Wordnet | | Citeseer | | Ogbn-Arxiv | | Mutag | | Molbace | |
|---|---|---|---|---|---|---|---|---|---|---|
| **Test** | | | | | | | | | | |
| **SD DiGress** | 0.65 | 0.50 | 0.81 | 0.61 | 0.88 | 0.86 | 0.78 | 0.77 | 0.79 | 0.96 |
| **SD GruM** | 0.52 | 0.97 | 0.78 | 0.69 | 0.48 | 0.21 | 0.33 | 0.65 | 0.54 | 0.48 |
| **SD GenStat** | 0.36 | 0.48 | 0.48 | 0.49 | 0.69 | 0.55 | 0.36 | 0.16 | 0.19 | 0.04 |
| **SD EDGE** | 0.53 | 0.20 | 0.92 | 0.83 | 0.55 | 0.14 | 0.16 | 0.53 | 0.21 | 0.10 |

Table 7: Performance of TOPOGEN for each attribute across all datasets. We report the mean absolute **Difference** from target attributes; lower is better (MAD↓ is the average over attributes).

| Attributes | Citeseer | WordNet | Mutag | Molbace | Ogbn-Arxiv |
|---|---|---|---|---|---|
| **Density** | 0.05 | 0.05 | 0.01 | 0.01 | 0.06 |
| **Edges** | 5.44 | 4.34 | 2.77 | 4.43 | 6.85 |
| **Nodes** | 1.89 | 3.73 | 0.00 | 2.98 | 3.68 |
| **NC** | 0.03 | 0.00 | 0.00 | 0.00 | 0.01 |
| **Avg Clust** | 0.16 | 0.17 | 0.03 | 0.07 | 0.19 |
| **CC** | 0.06 | 0.06 | 0.06 | 0.04 | 0.09 |
| **LB** | 4.16 | 7.11 | 2.33 | 6.89 | 4.14 |
| **Transitivity** | 0.13 | 0.12 | 0.04 | 0.09 | 0.10 |
| **EC** | 0.03 | 0.00 | 0.00 | 0.01 | 0.02 |
| **Cliques** | 6.08 | 3.73 | 2.00 | 4.16 | 7.19 |
| **TWMD** | 1.26 | 0.83 | 0.77 | 0.67 | 1.15 |
| **Diameter** | 1.71 | 1.53 | 2.77 | 3.47 | 2.22 |
| **MAD** | **1.71** | **1.80** | **1.00** | **1.90** | **2.14** |

## 5.7 NOVELTY OF GENERATED GRAPHS

To assess the novelty of the generated graphs, we quantified the extent to which the generated graphs are structurally distinct from those seen during training. Table 9 reports the fraction of graphs generated that are not isomorphic to any of the training graphs. This shows that our model generates structurally novel graphs that differ from the training distribution.

## 5.8 OUT OF DISTRIBUTION CONTROL ATTRIBUTES FOR GRAPH GENERATION

TOPOGEN uses graph reconstruction during training only to learn the conditional mapping from attributes to valid graph structures. At inference time, it relies only on the desired attributes to gen-

Table 8: Performance of TOPOGEN for each attribute for Ogbn-Arxiv dataset. Average mean absolute difference, MAD(↓), is the average of absolute mean error in satisfying target attributes.

| | WordNet | Citeseer | Ogbn-Arxiv | MUTAG | MOLBACE |
|---|---|---|---|---|---|
| | **MAD** | | | | |
| **GraphRNN** (You et al., 2018) | 3.26 | 5.05 | 4.80 | 1.71 | 3.81 |
| **GenStat** (Zahirnia et al., 2024) | 4.11 | 5.34 | 5.53 | 4.14 | 3.05 |
| **EDGE** (Chen et al., 2023) | 3.91 | 4.97 | 5.52 | 2.62 | 3.07 |
| **DiGress** (Vignac et al., 2023) | 5.23 | 6.63 | 6.67 | 5.39 | 9.96 |
| **GruM** (Jo et al., 2024) | 3.75 | 5.37 | 5.42 | 3.80 | 10.8 |
| **TOPOGEN** | **1.80** | **1.71** | **2.14** | **1.00** | **1.90** |

Table 9: Novelty(%) of generated graphs generated from TOPOGEN

| Dataset | Novelty (%) |
|---|---|
| Ogbn-Arxiv | 96.02 |
| Citeseer | 100.0 |
| MOLBACE | 100.0 |
| MUTAG | 100.0 |
| WordNet | 86.84 |

erate graphs. However, to assess generalization, we perform experiments to quantify if the trained model can accurately generate graphs from unseen out-of-distribution attributes—those that were not derived from the dataset used during training. To generate out-of-distribution attributes, we generated 25 random graphs through Barabási–Albert Barabási and Albert (1999) graph generation method and at the inference time used their attributes as the input to our model and report the generation error, SD (↓), in Table 10. It shows that the model maintains low generation error even on out-of-distribution attributes, demonstrating generalization beyond the training attribute distribution.

Table 10: Generation error, SD (↓) of TOPOGEN across datasets on out of distribution attributes.

| Model Trained on | SD (↓) |
|---|---|
| Citeseer | 0.37 |
| MOLBACE | 0.48 |
| MUTAG | 0.36 |
| Ogbn-Arxiv | 0.36 |
| WordNet | 0.32 |

## 5.9 ORDER-INVARIANCE

To analyze and mitigate the effect of order invariance, we re-ran our experiments using a consistent node ordering through BFS (as opposed to the random ordering in the paper) to reduce the overall number of sequences to be considered. The results in Table 12 show, averaged across all the datasets, that SD error slightly reduces when BFS node ordering is considered. This is because BFS preserves locality by placing structurally related nodes close to each other in the adjacency matrix. Such locality creates coherent spatial patterns that align with CNN kernels. This suggests that incorporating a more structured traversal order could improve stability by reducing sensitivity to arbitrary node permutations. In addition, our ablation study in Table 4 indicates that there is negligible increase in error when using GNNs as encoder.

## 5.10 MMD

Table 13, 15, 14 shows the MMD error across each attribute. Abbreviations NC (node connectivity), EC (edge connectivity), TWMD (tree width min degree), Avg Clust (average clustering), LB (number of local bridge), CC (Closeness Centrality).

Table 11: Performance of TOPOGEN using GNN as a graph encoder compared with CNN as an encoder. Average mean absolute difference, MAD(↓), is the average of absolute mean error in satisfying target attributes.

|  | WordNet | Citeseer | Ogbn-Arxiv | MUTAG | MOLBACE |
|---|---|---|---|---|---|
|  | **MAD** | | | | |
| **TOPOGEN w GNN** | 5.49 | 7.37 | 5.72 | 1.84 | 5.42 |
| **TOPOGEN** | **1.80** | **1.71** | **2.14** | **1.00** | **1.90** |

Table 12: Generation error showing comparison between BFS node ordering and random node ordering.

| Dataset | SD (BFS node ordering, ↓) | SD (random node ordering, ↓) |
|---|---|---|
| Arxiv | 0.26 | 0.40 |
| Citeseer | 0.21 | 0.27 |
| Molbace | 0.15 | 0.09 |
| Mutag | 0.14 | 0.10 |
| Wordnet | 0.18 | 0.44 |
| **Average** | **0.19** | **0.26** |

Table 13: MMD Results on Citeseer and MUTAG (lower is better, best in **bold**)

| Dataset |  | TOPOGEN | GenStat | EDGE | GruM | DiGress | GraphRNN |
|---|---|---|---|---|---|---|---|
| Citeseer | Density | **0.000** | **0.000** | 0.026 | 0.076 | 0.866 | **0.000** |
|  | Edges | **0.000** | **0.000** | 0.117 | **0.000** | 0.878 | 0.090 |
|  | Nodes | **0.000** | **0.000** | 0.085 | **0.000** | 0.910 | 0.015 |
|  | NC | 0.036 | **0.000** | **0.000** | **0.000** | 0.451 | 0.000 |
|  | Avg Clust | 0.262 | **0.025** | 0.397 | 0.170 | 0.880 | 0.401 |
|  | CC | **0.000** | **0.000** | **0.000** | 0.132 | 0.897 | 0.081 |
|  | LB | 0.274 | **0.071** | 0.314 | 0.168 | 0.973 | 0.100 |
|  | Transitivity | 0.139 | **0.137** | 0.233 | 0.076 | 0.845 | 0.370 |
|  | EC | 0.036 | 0.036 | **0.000** | **0.000** | 0.451 | 0.010 |
|  | LC | 0.074 | **0.000** | **0.000** | **0.000** | 0.790 | **0.000** |
|  | TD | 0.160 | **0.000** | 0.165 | **0.000** | 0.850 | 0.314 |
|  | Diameter | 0.239 | **0.000** | 0.209 | 0.202 | 1.116 | 0.146 |
| MUTAG | Density | **0.000** | **0.000** | **0.000** | 0.970 | 1.139 | 0.063 |
|  | Edges | **0.000** | 0.021 | 0.113 | 1.029 | 1.198 | 0.387 |
|  | Nodes | **0.000** | **0.000** | **0.000** | 0.980 | 1.181 | 0.242 |
|  | NC | **0.000** | **0.000** | **0.000** | **0.000** | **0.000** | **0.000** |
|  | Avg Clust | 0.594 | **0.000** | 1.158 | **0.000** | **0.000** | **0.000** |
|  | CC | 0.307 | 0.110 | **0.000** | 0.805 | 1.169 | 0.087 |
|  | LB | **0.000** | 0.089 | **0.000** | 0.127 | 0.771 | 0.365 |
|  | Transitivity | 0.580 | **0.000** | 1.185 | **0.000** | **0.000** | **0.000** |
|  | EC | **0.000** | **0.000** | **0.000** | **0.000** | **0.000** | **0.000** |
|  | LC | **0.000** | 0.021 | 0.179 | 1.029 | 1.198 | 0.387 |
|  | TD | 0.477 | 0.370 | 0.263 | 0.963 | 1.096 | 0.561 |
|  | Diameter | 0.411 | **0.000** | 0.191 | 0.718 | 1.069 | **0.000** |

Table 14: MMD Results on MOLBACE and WordNet (lower is better, best in **bold**)

| Dataset | | TOPOGEN | GenStat | EDGE | GruM | DiGress | GraphRNN |
|---------|---|---------|---------|------|------|---------|----------|
| | Density | 0.149 | **0.018** | 0.314 | 0.928 | 1.161 | 0.269 |
| | Edges | **0.000** | **0.000** | **0.000** | 1.206 | 1.204 | 0.454 |
| | Nodes | **0.000** | **0.000** | 0.099 | 1.194 | 1.190 | 0.371 |
| | NC | **0.000** | **0.000** | **0.000** | 0.604 | **0.000** | **0.000** |
| | Avg Clust | 0.724 | **0.000** | 0.806 | 0.065 | 0.067 | **0.000** |
| | CC | 0.297 | 0.030 | 0.577 | 0.960 | 1.223 | **0.000** |
| MOLBACE | LB | 0.182 | 0.100 | 0.372 | 0.989 | 0.894 | 0.318 |
| | Transitivity | 0.747 | **0.000** | 0.875 | 0.064 | 0.064 | **0.000** |
| | EC | **0.000** | **0.000** | **0.000** | 0.604 | **0.000** | **0.000** |
| | LC | 0.074 | **0.000** | 0.305 | 1.164 | 1.201 | 0.450 |
| | TWMD | 0.592 | **0.000** | 0.714 | 1.122 | 1.099 | 0.544 |
| | Diameter | 0.290 | 0.036 | 0.568 | 1.192 | 1.197 | **0.000** |
| | Density | 0.194 | **0.000** | 0.144 | 0.407 | 1.119 | 0.224 |
| | Edges | 0.108 | **0.000** | 0.125 | 0.382 | 0.899 | 0.224 |
| | Nodes | 0.120 | **0.000** | 0.130 | 0.414 | 0.894 | 0.214 |
| | NC | 0.002 | 0.000 | 0.004 | 0.065 | 0.026 | **0.000** |
| | Avg Clust | 0.602 | **0.000** | 0.038 | 0.169 | 0.216 | 0.153 |
| | CC | 0.380 | **0.000** | 0.110 | 0.345 | 1.045 | 0.164 |
| WordNet | LB | 0.283 | **0.000** | 0.129 | 0.434 | 0.885 | 0.179 |
| | Transitivity | 0.679 | **0.000** | 0.080 | 0.203 | 0.180 | 0.128 |
| | EC | 0.004 | 0.000 | 0.004 | 0.065 | 0.026 | **0.000** |
| | LC | 0.085 | **0.000** | 0.134 | 0.403 | 0.888 | 0.209 |
| | TWMD | 0.613 | **0.000** | 0.008 | 0.143 | 0.324 | 0.207 |
| | Diameter | 0.549 | 0.014 | 0.072 | 0.098 | 0.612 | **0.017** |

Table 15: MMD Results on ArXiv (lower is better, best in **bold**)

| Dataset | | TOPOGEN | GenStat | EDGE | GruM | DIiress | GraphRNN |
|---------|---|---------|---------|------|------|---------|----------|
| | Density | 0.069 | **0.012** | 0.092 | 0.224 | 1.004 | 0.073 |
| | Edges | 0.046 | **0.011** | 0.091 | 0.190 | 0.834 | 0.137 |
| | Nodes | 0.040 | 0.019 | 0.072 | **0.000** | 0.905 | 0.096 |
| | NC | **0.000** | **0.000** | 0.001 | 0.009 | 0.158 | 0.003 |
| | Avg Clust | 0.476 | **0.000** | 0.198 | 0.852 | 0.850 | 0.355 |
| | CC | 0.343 | **0.000** | 0.150 | 0.328 | 1.030 | 0.097 |
| Ogbn-Arxiv | LB | 0.169 | **0.000** | 0.213 | 0.397 | 0.872 | 0.061 |
| | Transitivity | 0.100 | **0.000** | 0.136 | 0.772 | 0.771 | 0.287 |
| | EC | **0.000** | **0.000** | 0.006 | 0.011 | 0.158 | 0.005 |
| | LC | 0.067 | 0.018 | 0.075 | 0.066 | 0.839 | **0.077** |
| | TWMD | 0.125 | 0.023 | 0.071 | 0.696 | 0.747 | 0.231 |
| | Diameter | 0.598 | 0.014 | 0.324 | **0.017** | 0.990 | 0.205 |

