# OpenReview forum: "Fine-Grained Graph Generation through Latent Mixture Scheduling"
_ICLR.cc/2026/Conference — Submitted to ICLR 2026_

### Official Review · Reviewer_3x2H · 2025-10-17

**Soundness:** 2
**Presentation:** 3
**Contribution:** 2
**Rating:** 4
**Confidence:** 4

**Summary:**

This paper presents TOPOGEN, a conditional variational autoencoder designed for attribute-controlled graph generation. The model introduces a Mixture-Scheduler that gradually integrates structural and attribute-based latent representations during training, aiming to better align generated graphs with desired topological properties. The method supports fine-grained control over various structural attributes such as density, transitivity, and connectivity. The paper is generally well-written and easy to follow.

However, the work lacks strong theoretical grounding, and the empirical evidence supporting the model’s effectiveness is limited.

1. In particular, it is unclear why the authors constructed new benchmark datasets (Section 3, Datasets) instead of evaluating their model on standard benchmarks such as GRID, LOBSTER, or PROTEIN, which are commonly used in prior studies like GraphRNN and GenStat.
2. There is also insufficient evidence regarding the diversity of the generated graphs. Since TOPOGEN conditions generation on attribute vectors, additional analysis is needed to confirm that the model is not simply memorizing graphs from training data. Established metrics such as F1-PR [1,2] could have been included to better evaluate the diversity and novelty of generated graphs.
3. Furthermore, the results reported in Table 2 are not fully consistent with the qualitative examples shown in Table 6. For instance, the graphs generated by DiGress appear visually similar across all datasets, often small tree-like structures with very few nodes, which do not match the ground truth samples. Yet, according to the GED metric, DiGress performs best among all models on Citeseer and Ogbn-Arxiv. This discrepancy raises concerns about the suitability of evaluation metric.
4. Finally, while the Mixture-Scheduler is clearly described mathematically, the paper does not formally connect it to the variational inference objective, how it modifies or influences the ELBO formulation.

[1]  Rylee Thompson, Boris Knyazev, Elahe Ghalebi, Jungtaek Kim, and Graham W Taylor. On evaluation metrics for graph generative models. In International Conference on Learning Representations, 2022

[2] Shirzad, Hamed, Kaveh Hassani, and Danica J. Sutherland. "Evaluating graph generative models with contrastively learned features." Advances in Neural Information Processing Systems 35 (2022): 7783-7795.

**Strengths:**

The paper is easy to read and well-organized, making the proposed ideas and model design straightforward to follow.
The introduction of the Mixture-Scheduler is an interesting idea that gradually balances structural and attribute-driven latent representations.
TOPOGEN supports precise control over diverse graph-level attributes such as density, connectivity, and transitivity, which is a step beyond coarse-grained control in existing models.

**Weaknesses:**

Scalability and Generalization Limitations
Limited Empirical Validation
Lack of Theoretical Grounding

**Questions:**

How is the attribute vector c sampled during inference? Is it derived from sampled graphs from the training or test sets?

---

> ### Author Response · Authors · 2025-11-14
> **Author's Response**
>
> We thank the reviewer for their comments.
>
> **Q.** How is the attribute vector c sampled during inference? Is it derived from sampled graphs from the training or test sets?
>
> **A.** During inference, the attribute vector (c) is given for each graph in the test set (it is not derived or sampled).This setting allows us to directly evaluate each model's ability to generate graphs that satisfy the specified target attributes. We note that, at inference time, our model generates graphs using only the prior $p_{\theta}$  and the likelihood $p_{\psi}$, conditioned solely on the given attribute vector c.
>
> **Q.** In particular, it is unclear why the authors constructed new benchmark datasets (Section 3, Datasets) instead of evaluating their model on standard benchmarks such as GRID, LOBSTER, or PROTEIN, which are commonly used in prior studies like GraphRNN and GenStat.
>
> **A.** Our goal is to demonstrate that fine-grained topological control is broadly applicable across multiple real-world domains. While we didn’t experiment with the above domain-specific datasets, we included datasets spanning academic (Ogbn-Arxiv, Citeseer), linguistic (Wordnet) and biological domains (MUTAG, MOLBACE) graphs. We believe this diversity allows us to fairly evaluate models across different structural patterns and domains. In addition, we note that there is no single standardized benchmark that is consistently used across studies—while models are often applicable across datasets, different works often adopt different datasets depending on their task focus.
>
>
> **Q.** There is also insufficient evidence regarding the diversity of the generated graphs. Since TOPOGEN conditions generation on attribute vectors, additional analysis is needed to confirm that the model is not simply memorizing graphs from training data. Established metrics such as F1-PR [1,2] could have been included to better evaluate the diversity and novelty of generated graphs.
>
> **A.** Our model generates graphs solely from attribute vectors and does not access training graphs during inference, which prevents memorization by design. To assess the novelty of the generated graphs, we quantified the extent to which the generated graphs are structurally distinct from those seen during training. Specifically, the following Table reports the fraction of generated graphs that are not isomorphic to any of the training graphs:
>
> | Dataset  | Novelty (%) |
> |----------|-------------|
> | Ogbn-Arxiv    | 96.02       |
> | Citeseer | 100.0       |
> | MOLBACE  | 100.0       |
> | MUTAG    | 100.0       |
> | Wordnet  | 86.84       |
>
> **Q.** Furthermore, the results reported in Table 2 are not fully consistent with the qualitative examples shown in Table 6. For instance, the graphs generated by DiGress appear visually similar across all datasets, often small tree-like structures with very few nodes, which do not match the ground truth samples. Yet, according to the GED metric, DiGress performs best among all models on Citeseer and Ogbn-Arxiv. This discrepancy raises concerns about the suitability of evaluation metric.
>
> **A.** Table 6 only presents a few qualitative examples while Table 2 reports overall quantitative performance across the entire test set. The examples are meant to illustrate representative cases, not to reflect full dataset-level trends.
>
> **Q.** Finally, while the Mixture-Scheduler is clearly described mathematically, the paper does not formally connect it to the variational inference objective, how it modifies or influences the ELBO formulation.
>
> **A.** The Mixture-Scheduler acts as a latent-space interpolation mechanism that adaptively balances the prior p and posterior q during training. The scheduler modifies the KL term by gradually aligning these two distributions through the weighted inclusion factor $\beta$. This results in a smoother posterior–prior matching and more stable optimization. We will clarify this connection to the ELBO in the revised version for completeness.

---

> > ### Author Response · Authors · 2025-11-28
> >
> > Dear Reviewer 3x2H,
> >
> > We have uploaded a revised version of our paper that includes the details and clarifications  mentioned in our response, highlighted in blue. Please let us know if the below additions address your concerns. Below are the update details :
> >
> >  **Section 5.7 in Appendix: Novelty of Generated Graphs, Table 9:** Table 9 reports the fraction of generated graphs that are non-isomorphic to any training graph. This shows that our model generates structurally novel graphs that differ from the training distribution.

---

### Official Review · Reviewer_A1PX · 2025-10-28

**Soundness:** 3
**Presentation:** 3
**Contribution:** 3
**Rating:** 4
**Confidence:** 3

**Summary:**

This paper introduce TopoGen, a graph VAE for fine-grained structural control (e.g., controlling density, connectivity, cliques, clustering, etc) in graph generation. Compared to standard conditioned graph VAE, the core innovation of TopoGen is that, it introduce a novel training curriculum, where at the early training stage, the model is fed with both the graph and the target graph structural attributes; however, as the training move on, the contribution of the graph to the VAE's hidden representation is gradually decayed to zero. Finally, the model is able to reconstruct the target graph distribution via VAE with solely attribute information.

**Strengths:**

- This paper propose a novel Mix-Scheduler training curriculum to enable graph VAE to perform conditioned graph generation based on curriculum learning.
- This paper evaluates the effectiveness of the proposed TopoGen on several conditioned graph generation benchmarks.
- The design of TopoGen is simple yet effective.

**Weaknesses:**

In general, this paper is well-written, and the empricial results are convincing. I will consider raising my score after the follow concerns are addressed:

**Concern 1. Does TopoGen generates novel graphs?**

To validate the novelty and validity of the generated graphs, I recommend the authors to add experiemnts to compare the generated graph data and the training dataset to check if the generated graphs are indeed novel.


**Concern 2. Network Architecture of TopoGen**

To improve the clarity of the model, I recommend the authors to provide a detail illustration on the network structure of each TopoGen modules. The parameter size and FLOPs are also recommended to be included in the manuscript.


**Concern 3. Can TopoGen control multiple attributes simultaneously?**

I am curious about the performance of TopoGen when generating graphs with multiple attributes specified simultaneously (e.g., generating graphs with certain density, certain node and edge connectivity, etc). Will the performance degraded when the specified attributes are rare in the training dataset? Or it can achieve comparable performance to the single attribute generation task?

**Questions:**

See above.

---

> ### Author Response · Authors · 2025-11-14
> **Author's Response**
>
> We thank the reviewer for their comments.
>
> **Q.** Does TopoGen generate novel graphs? To validate the novelty and validity of the generated graphs, I recommend the authors to add experiments to compare the generated graph data and the training dataset to check if the generated graphs are indeed novel.
>
> **A.** To assess the novelty of the generated graphs, we quantified the extent to which the generated graphs are structurally distinct from those seen during training. Specifically, the following Table reports the fraction of generated graphs that are not isomorphic to any of the training graphs:
>
> | Dataset  | Novelty (%) |
> |----------|-------------|
> | Arxiv    | 96.02       |
> | Citeseer | 100.0       |
> | Molbace  | 100.0       |
> | Mutag    | 100.0       |
> | Wordnet  | 86.84       |
>
> **Q.** Network Architecture of TopoGen: To improve the clarity of the model, I recommend the authors to provide a detail illustration on the network structure of each TopoGen modules. The parameter size and FLOPs are also recommended to be included in the manuscript.
>
> **A.**
> FLOPs: 260M (graph encoder) + 260M (graph decoder)+ 231M (attribute encoder) + 231M (attribute decoder)
>
> Parameters: 116M (100k (CNN) + 115M (MLP) with hidden dim size=1024)
>
> In addition, attribute encoder and decoder are two-layer MLP each. We consider a maximum number of 1000 training iterations for Citeseer and 200 iterations for other datasets, which is sufficiently large for convergence. We consider a batch-size of 1,028 and run all our experiments on a single A100 40GB GPU.
>
>
> **Q.** Can TopoGen control multiple attributes simultaneously? I am curious about the performance of TopoGen when generating graphs with multiple attributes specified simultaneously (e.g., generating graphs with certain density, certain node and edge connectivity, etc). Will the performance degraded when the specified attributes are rare in the training dataset? Or it can achieve comparable performance to the single attribute generation task?
>
> **A.** Yes, TopoGen is designed to and evaluated in controlling multiple attributes simultaneously. In the experiments, the model takes an attribute control “vector” as input, which represents the desired attribute values. Table 2 shows the results. In addition, Figure 6 shows the change in error when any one attribute is masked (set to zero). This can be considered a rare attribute value. Results show that TOPOGEN remains resilient, often generating accurate graphs despite noisy/rare attributes. The largest error increases occur when edges, local bridges, or cliques are masked, confirming their critical role in structural fidelity, whereas masking clustering coefficient, transitivity, or diameter yields only minor changes, indicating they refine finer structural details.
>
> **C.** In general, this paper is well-written, and the empricial results are convincing. I will consider raising my score after the follow concerns are addressed:
>
> **A.** We thank the reviewer for their consideration.

---

> > ### Comment · Reviewer_A1PX · 2025-11-24
> >
> > Dear authors, I have read your response. I believe the revising the manuscript according to the added experiments and analysis can improve the paper.

---

> > > ### Author Response · Authors · 2025-11-24
> > >
> > > We thank the reviewer for taking the time to read our response. We would like to clarify that all specific comments raised in your original review have been directly addressed: we reported the statistics of novel graphs, provided FLOP and parameter counts, and clarified that the model already supports controlling multiple attributes simultaneously. We believe our response clearly addressed the reviewer’s earlier comments. However, since the overall evaluation and score were not updated, and the reviewer’s recent comment (“I believe revising the manuscript according to the added experiments and analysis can improve the paper”) does not specify any remaining issues, we respectfully request clarification on which aspects--if any--from the original comments the reviewer still finds insufficiently addressed.

---

> ### Comment · Reviewer_A1PX · 2025-11-27
>
> Dear authors, you are recommended to update your manuscript accordingly.

---

> > ### Author Response · Authors · 2025-11-28
> >
> > Dear Reviewer A1PX,
> >
> > We have uploaded a revised version of our paper that includes the details and clarifications  mentioned in our response, highlighted in blue. Please let us know if the below additions address your concerns. Below are the update details :
> >
> >  1. **Section 5.7 in Appendix: Novelty of Generated Graphs, Table 9:** Table 9 reports the fraction of generated graphs that are non-isomorphic to any training graph. This shows that our model generates structurally novel graphs that differ from the training distribution.
> >
> >  2. We have updated **Section 5.3 in Appendix** with details about model parameter size and FLOPs.

---

> > > ### Comment · Reviewer_A1PX · 2025-11-28
> > >
> > > Dear Authors, given the revised manuscript, I would like to raise my score from 4 to 6. But for now, the rebuttal information button is not activated.

---

### Official Review · Reviewer_zW3B · 2025-10-30

**Soundness:** 3
**Presentation:** 3
**Contribution:** 2
**Rating:** 6
**Confidence:** 3

**Summary:**

The paper proposes TOPOGEN, a conditional VAE for fine-grained, attribute-controlled graph generation. During training, the model conditions on both the graph’s adjacency matrix and a vector of topological attributes; during inference, it generates graphs from attributes alone.

**Strengths:**

1. The paper targets fine-grained, attribute-controlled graph generation where, at test time, only a target attribute vector is provided and the model must synthesize a graph  matching those properties. That “attributes-only inference” setting is less explored than standard conditional generation (which often still uses partial structure), and the paper treats it as a first-class goal.
2. The paper doesn’t just tune a single property; it conditions on a set of structural descriptors and tests across diverse graph families . That variety increases confidence that the approach is not tailored to a single metric or domain.
3. By conditioning on interpretable attributes, the model creates a direct interface between human-specified constraints and learned generative structure, which is useful for scientific and engineering workflows where knobs like clustering, path length, or degree profiles are the design targets.

**Weaknesses:**

1. TOPOGEN’s structural encoder is a CNN over adjacency matrices; ablations even note that replacing CNN with a GNN hurts performance, which the paper attributes to over-smoothing. But CNNs on raw adjacencies are not permutation-invariant, so generation quality could depend on node ordering.

**Questions:**

See above

---

> ### Author Response · Authors · 2025-11-14
> **Author's Response**
>
> We thank the reviewer for their comments.
>
> **Q.** TOPOGEN’s structural encoder is a CNN over adjacency matrices; ablations even note that replacing CNN with a GNN hurts performance, which the paper attributes to over-smoothing. But CNNs on raw adjacencies are not permutation-invariant, so generation quality could depend on node ordering.
>
> **A.**  As mentioned by the reviewer, CNNs over adjacency matrices are not permutation-invariant. To analyze and mitigate the effect of this limitation, we re-ran our experiments using a consistent node ordering through BFS (as opposed to the random ordering in the paper) to reduce the overall number of sequences to be considered. The results show, averaged across all the datasets,  that SD error slightly reduces when BFS node ordering is considered.
>
> | Dataset  | SD (BFS node ordering, ↓) | SD (random node ordering, ↓) |
> |----------|---------------------------|-------------------------------|
> | Ogbn-Arxiv    | 0.26                      | 0.40                          |
> | Citeseer | 0.21                      | 0.27                          |
> | MOLBACE  | 0.15                      | 0.09                          |
> | MUTAG    | 0.14                      | 0.10                          |
> | Wordnet  | 0.18                      | 0.44                          |
> | **Average** | **0.19**               | **0.26**                      |
>
> In addition, our ablation study in Table 4 indicates that there is negligible increase in error when using GNNs as encoder.

---

> > ### Comment · Reviewer_zW3B · 2025-11-25
> >
> > Thanks authors for the additional experiments. I will keep my accept score.

---

> > > ### Author Response · Authors · 2025-11-28
> > >
> > > Dear Reviewer zW3B,
> > >
> > > We have uploaded a revised version of our paper that includes the details and clarifications  mentioned in our response, highlighted in blue. Please let us know if the below additions address your concerns. Below are the update details :
> > >
> > > **Section 5.9: Order-Invariance, Table 12:** The results in Table12 show, averaged across all the datasets, SD error slightly reduces when BFS node ordering is considered. This is because BFS preserves locality by placing structurally related nodes close to each other in the adjacency matrix. Such locality creates coherent spatial patterns that align with CNN kernels. This suggests that incorporating a more structured traversal order could improve stability by reducing sensitivity to arbitrary node permutations.

---

### Official Review · Reviewer_dZzG · 2025-11-01

**Soundness:** 2
**Presentation:** 3
**Contribution:** 2
**Rating:** 4
**Confidence:** 3

**Summary:**

This paper introduces TOPOGEN, a new framework for fine-grained controllable graph generation. The motivation is that most existing graph generators can only handle coarse attributes (like node or edge counts), but they fail to precisely satisfy more detailed topological constraints such as density, transitivity, or connectivity.

To address this, the authors build a conditional variational autoencoder (CVAE) and propose a core component called the MIXTURE-SCHEDULER. This scheduler dynamically blends the latent representations learned from both the graph structure (adjacency matrix) and the attribute vector. Over the course of training, it gradually shifts the model from being structure-driven to attribute-driven, which helps maintain structural realism while improving controllability.

**Strengths:**

Clear motivation and importance.
The paper addresses a meaningful gap that most existing graph generators only allow coarse control (like node or edge counts), while this work focuses on fine-grained topological controllability. The problem is well-motivated and relevant to applications such as molecule design and knowledge graphs.
2. Methodologically sound and conceptually novel.
The proposed TOPOGEN model builds on a conditional VAE and introduces a Mixture-Scheduler that gradually blends structure and attribute latent spaces during training. This design is intuitive, novel, and provides a clear mechanism to balance structure realism and control accuracy.
3. Empirical evidence supports the claims.
Ablation studies clearly show the contribution of each module (scheduler, distance term, attribute reconstruction), strengthening the technical soundness of the work.

**Weaknesses:**

1. Dataset setup leans toward reconstruction.
Most graphs are extracted subgraphs (k-hop) from larger networks, making the task more like reconstruction than true open-ended generation, which limits generalization testing.
2. Lack of standard generation benchmarks.
Common datasets such as QM9, ZINC, or synthetic graphs are missing, which makes it difficult to compare directly with recent diffusion-based or energy-based models
3. Metrics mainly cover structure similarity.
While GED and SD are appropriate, they do not directly measure controllability (e.g., attribute accuracy) or diversity. Adding such metrics would provide a more complete evaluation.

**Questions:**

1. The datasets used (WordNet, Citeseer, Arxiv, MUTAG, MOLBACE) are quite diverse, but most are derived subgraphs rather than standard graph-generation benchmarks. Could you explain why these datasets were preferred over commonly used benchmarks such as QM9, ZINC, or synthetic graphs? How do these datasets help demonstrate fine-grained controllability compared to molecular or synthetic graph families, where topology can be more precisely measured?
2. The main evaluation metrics (GED and SD) capture structural similarity, but not controllability. Have you considered any other evaluation metric for controllability?
3. Since this work deals with graph generation, evaluating the novelty of generated graphs is particularly important — especially for domains like drug discovery, where creating genuinely new molecular structures is a key objective. Have you considered including novelty/diversity metrics to better assess generation quality?

---

> ### Author Response · Authors · 2025-11-15
>
> We thank the reviewer for their comments.
>
> **Q.** Dataset setup leans toward reconstruction. Most graphs are extracted subgraphs (k-hop) from larger networks, making task like reconstruction than true open-ended generation.
>
> **A.** TopoGen uses graph reconstruction during training only to learn the conditional mapping from attributes to valid graph structures. At the inference time, it only relies on desired attributes to generate graphs. However, to assess generalization beyond local reconstruction, we also performed experiments to quantify if the trained model can accurately generate graphs from unseen attributes—those that were not used during training. The following Table reports the results:
> |Model Trained on|SD(↓)|
> |-|-|
> |Citeseer|0.37|
> |MOLBACE|0.48|
> |MUTAG|0.36|
> |Ogbn-Arxiv|0.36|
> |WordNet|0.32|
>
> **Q.** The datasets used (WordNet, Citeseer, Arxiv, MUTAG, MOLBACE) are quite diverse, but are derived subgraphs rather than standard graph-generation benchmarks. Could you explain why these datasets were preferred over common benchmarks such as QM9, ZINC, synthetic graphs? How do these datasets demonstrate fine-grained controllability for molecular/synthetic graph, where topology can be precisely measured?
>
> **A.** QM9 and ZINC datasets primarily represent molecular graphs from the chemical domain. Our goal is to demonstrate that fine-grained topological control is broadly applicable across multiple real-world domains. While we didn’t experiment with the above domain-specific datasets, we included datasets spanning academic (Ogbn-Arxiv, Citeseer), linguistic (Wordnet) and biological domains (MUTAG, MOLBACE) graphs. We believe this diversity allows us to fairly evaluate our model’s generalization ability across different structural patterns and domains. In addition, our framework is designed around graph-theoretic attributes, which are domain agnostic and can generalize to any type of graphs. We note that there is no single standardized benchmark that is consistently used across studies—while models are often applicable across datasets, different works often adopt different datasets depending on their task focus. Finally, we compared our model against strong diffusion based models (DiGress, EDGE, GruM) and observed consistent improvement across all datasets.
>
> **Q.** While reported metrics GED and SD are appropriate and  capture structural similarity, they do not directly measure controllability e.g., attribute accuracy/diversity. Have you considered any other evaluation metric for controllability?
>
> **A.** Yes, Section 5.5 in Appendix: Table 7 reports the mean absolute error (MAD↓) between the ground truth and predicted values for each control attribute. This metric directly measures how accurately models satisfy attribute values. In addition, Table 8 reports the aggregation of these results using the MAD metric for all models.
>
> Table 7: Performance of the model for each attribute across all datasets
> |Attributes|Citeseer|WordNet|Mutag|Molbace|Ogbn-Arxiv|
> |-|-|-|-|-|-|
> |Density|0.05|0.05|0.01|0.01|0.06|
> |Edges|5.44|4.34|2.77|4.43|6.85|
> |Nodes|1.89|3.73|0.00|2.98|3.68|
> |NC|0.03|0.00|0.00|0.00|0.01|
> |AvgClust|0.16|0.17|0.03|0.07|0.19|
> |CC|0.06|0.06|0.06|0.04|0.09|
> |LB|4.16|7.11|2.33|6.89|4.14|
> |Transitivity|0.13|0.12|0.04|0.09|0.10|
> |EC|0.03|0.00|0.00|0.01|0.02|
> |Cliques|6.08|3.73|2.00|4.16|7.19|
> |TWMD|1.26|0.83|0.77|0.67|1.15|
> |Diameter|1.71|1.53|2.77|3.47|2.22|
> |**MAD**|**1.71**|**1.80**|**1.00**|**1.90**|**2.14**|
>
> Table 8 Performance of the Model for Each Attribute (MAD ↓)
> |Model|WordNet|Citeseer|Ogbn-Arxiv|MUTAG|MOLBACE|
> |-|-|-|-|-|-|
> |GraphRNN|3.26|5.05|4.80|1.71|3.81|
> |GenStat|4.11|5.34|5.53|4.14|3.05|
> |EDGE|3.91|4.97|5.52|2.62|3.07|
> |DiGress|5.23|6.63|6.67|5.39|9.96|
> |GruM|3.75|5.37|5.42|3.80|10.8|
> |**TOPOGEN**|**1.80**|**1.71**|**2.14**|**1.00**|**1.90**|
>
> **Q.** This work deals with graph generation, where the novelty of generated graphs is important. Have you considered including novelty metrics?
>
> **A.** To assess the novelty of the generated graphs and generalization beyond local reconstruction, we quantified the extent to which the generated graphs are structurally distinct from those seen during training. Specifically, the following Table reports the fraction of generated graphs that are not isomorphic to any of the training graphs:
> |Dataset|Novelty(%)|
> |-|-|
> |Arxiv|96.02|
> |Citeseer|100.0|
> |Molbace|100.0|
> |Mutag|100.0|
> |Wordnet|86.84|
>
> **Q.** They fail to precisely satisfy more detailed topological constraints such as density, transitivity, or connectivity.
>
> **A.** As shown in Table 7 (Section 5.5), the MAD (↓) for density, transitivity and closeness centrality attributes are among the lowest across all datasets, indicating that the model accurately satisfies those attributes. Furthermore, our ablation analysis in Fig 3 indicates that these attributes play a crucial role in precise generation as the generation error increases as any one of them is removed from the list of control attributes.

---

> > ### Author Response · Authors · 2025-11-28
> >
> > Dear Reviewer dZzG,
> >
> > We have uploaded a revised version of our paper that includes the details and clarifications  mentioned in our response, highlighted in blue. Please let us know if the below additions address your concerns. Below are the update details :
> >
> > 1. **Section 5.8 in Appendix: Out-of-distribution Control Attributes for Graph Generation, Table 10:** We tested our model with the attributes of 25 random graphs (obtained through Barabási–Albert method) and reported its generation error SD (↓)  in Table 10. The results show that the model maintains low generation error even on out-of-distribution attributes, which demonstrates generalization beyond the training attribute distribution.
> >
> > 2. **Section 5.7 in Appendix: Novelty of Generated Graphs, Table 9:** Table 9 reports the fraction of generated graphs that are non-isomorphic to any training graph. This shows that our model generates structurally novel graphs that differ from the training distribution.

---

### Meta-Review · Area_Chair_C7kd · 2026-01-06

**Summary:**

The paper introduce TOPOGEN, a new method based on conditional variational autoencoder (CVAE) for graph generation,  controlled graph generation utilising the fine-grained topological attributes. The training phase includes both graph adjacency matrix and the given core topological attributes, while the generation phase only uses the core attributes to reconstruct target graphs in distribution. The key component involves what is called Mixture-scheduler that gradually integrates the structure and attribute representation. Experiment results are conducted in benchmarking dataset for graph generation.

**Reviewer Concerns:**

TOPOGEN architecture unclear. The point is address by additional clarification on parameters, layer structures, and attribute encoders.

Generating novel graphs or graph diversity concerns. The concern is partially addressed in discussion by providing a list of experimental results for novelty using a notion described as "structurally distinct" from seen training graphs. It remains unclear how the structural distinction is defined, e.g. computed via hamming distances, Weisfeiler Leman graph isomorphism, or other specific metric.

The comment from reviewer dZzG on satisfying detailed topological constraints such as density, transitivity or connectivity. This point was not explicitly addressed.

**Reviewer Scores:**

may not change

---

### Decision · Program_Chairs · 2026-01-26

Reject